# Emerging Biomarkers and Innovative Therapeutic Strategies in Diabetic Kidney Disease: A Pathway to Precision Medicine

**DOI:** 10.3390/diagnostics15080973

**Published:** 2025-04-11

**Authors:** Sahana Shetty, Renuka Suvarna, Avivar Awasthi, Mohan V. Bhojaraja, Joseph M. Pappachan

**Affiliations:** 1Department of Endocrinology, Kasturba Medical College, Manipal, Manipal Academy of Higher Education, Manipal 576104, India; sahana.shetty@manipal.edu (S.S.); renuka.kmc@manipal.edu (R.S.); awasthi.avivar@manipal.edu (A.A.); 2Department of Nephrology, Kasturba Medical College, Manipal, Manipal Academy of Higher Education, Manipal 576104, India; mohan.vb@manipal.edu; 3Faculty of Science, Manchester Metropolitan University, Manchester M15 6BH, UK

**Keywords:** diabetic kidney disease, diabetic nephropathy, diabetes, biomarkers, therapeutic strategies

## Abstract

Diabetes mellitus (DM) has emerged as the most common cause of chronic kidney disease (CKD) and end-stage renal disease (ESRD) globally in recent years. Diabetic nephropathy (DN), or diabetic kidney disease (DKD) that occurs as a direct consequence of DM, has complex pathophysiological mechanisms, such as various inflammatory processes and genetic and epigenetic factors, often accentuated by comorbid illnesses like hypertension and dyslipidemia. Therefore, management of DKD involves targeting these etio-pathological processes. Various medications with remarkable disease modifying properties have been introduced for treatment of DN in recent years. We update the current and future diagnostic and therapeutic landscapes against DKD in this article.

## 1. Introduction

Diabetes mellitus (DM) has become the most common cause of chronic kidney disease (CKD), contributing to one-third of cases globally [1,2]. It has been shown that CKD substantially increases morbidity and mortality worldwide in diabetic patients regardless of the income status of the population studied, especially in developing economies [3]. The prevalence of diabetic kidney disease (DKD) is expected to increase further considering the relentless rise in the incidence of both type 1 DM (T1DM) and type 2 DM (T2DM) all over the world in recent years. Diabetic nephropathy (DN; the alternative terminology often used for DKD) is one of the very serious complications of DM. DKD has now emerged as the most common cause of end-stage renal disease (ESRD) across the globe.

DKD is diagnosed when patients develop CKD as a direct consequence of DM without other contributing illnesses, such as hypertension, renovascular disease, and secondary glomerular disorders [4]. Although the disease is highly prevalent and identified as the cause of increased incidence of end-stage kidney disease (ESKD), the pathobiological mechanisms for DKD remain incompletely understood [5]. Marked structural and functional alterations of the kidney are identified in patients with DKD, and newer therapeutic strategies are being explored in their attempts to stop the disease onset and progression. This review provides an insight into the recent advances of the pathobiology of DKD, evaluation with a specific focus on novel biomarkers, and emerging disease-modifying approaches for this enigmatic disease.

## 2. Pathobiology of Diabetic Nephropathy

Glomerular hyperfiltration and intraglomerular hypertension have long been key hemodynamic factors in DKD progression [6]. However, DKD pathophysiology is now recognized as multifactorial, involving metabolic, inflammatory, hemodynamic, and fibrotic pathways [7]. Renin–angiotensin–aldosterone system (RAAS) activation, advanced glycation end-product (AGE) accumulation, and epithelial–mesenchymal transition (EMT) contribute to cellular stress, inflammation, apoptosis, and autophagy in DKD [8]. Microvascular damage leads to tubulointerstitial fibrosis and glomerular sclerosis, driven by capillary injury and Glomerular Basement Membrane (GBM) alterations [9,10]. Though highly invasive, renal biopsy and histological analysis including electron microscopy help us to understand some of these pathobiological aspects.

Angiotensin II affects mesangial cells and promotes the expression of profibrotic and proinflammatory mediators [11]. They also activate the inflammatory response with increased production of cytokines like transforming growth factor (TGF)-β and reactive oxygen species, which also mediate the glomerular damage and apoptosis of podocytes, along with microvascular, interstitial, and tubular damage [12,13].

Hyperglycemia-induced biochemical reactions contribute to kidney injury by promoting fibrosis and inflammation. Disrupted intracellular glucose metabolism activates inflammatory pathways, leading to AGEs and reactive oxygen species production, along with protein kinase C and JAK activation [14]. AGE-RAGE interactions in podocytes and endothelial cells trigger inflammation via pyrin domain-containing inflammasomes, promoting IL-1β and IL-18 activation through NF-kB and nucleotide-binding oligomerization domain-like receptors. Additionally, serum amyloid A, another RAGE activator, enhances inflammatory gene expression, further amplifying inflammation [15].

Genetic factors also predispose one to the development of DKD, although determining the genetic contributions to the severity of DKD poses substantial challenges. Notably, people with T1DM and high glycated hemoglobin (HbA1c) levels showed the greatest protection against DKD with this *COL4A3* variation. In individuals with either T1DM or T2DM, this *COL4A3* variation is also linked to reduced glomerulosclerosis and GBM thickness [16]. The *COL4A3* variant (rs55703767) has been associated with protection against albuminuria in T1DM patients, suggesting a potential role in maintaining kidney function. Additional variants such as *DDR1*, *COLEC11*, and *BMP7* are also linked to collagen abnormalities and renal fibrosis. *APOL1G1/G2* alleles are associated with non-diabetic CKD, and emerging evidence indicates that they may also contribute to the accelerated progression of DKD in African Americans. Microalbuminuria was strongly correlated with a variation in the *GABRR1* gene (rs9942471) in a European population of patients with T2DM [17]. Apart from genetic analysis, epigenetic modifications like DNA methylation impact the genotype effect on DKD. Specifically, different alterations in cytosine methylation govern the immune system and inflammation, involving macrophage clearance of apoptotic cells. Establishing a genetic correlation with DKD is crucial for deciphering the susceptibility to the disease and possible targets for treatment [18].

Diabetes Control and Complications Trial (DCCT) and its follow-up, the Epidemiology of Diabetes Interventions and Complications (EDIC) studies, have shown that intensive glycemic control plays a crucial role in reducing the risk of early-stage DKD in individuals with type 1 diabetes, highlighting its importance in preventing microvascular complications [19]. Long-lasting epigenetic changes, such as histone methylation or acetylation, brought by long-term diabetes led to the activation of profibrotic and proinflammatory genes, resulting in progressive changes in DKD even with intense glycemic control [15].

## 3. Structural Changes in the Kidney

The pathogenesis of DKD is complex and can affect the entire structure of the kidney, including the glomerulus, tubules, and interstitium. The morphological alterations associated with DKD comprise tubular hypertrophy, podocyte dysfunction, mesangial expansion, thickening of the basement membrane, and tubulointerstitial fibrosis. These changes advance with progression in kidney lesions which increases oxidative stress, fibrogenesis, inflammation, and cell death [20].

Histologically, DKD is characterized by glomerular and tubular basement membrane thickening, mesangial matrix expansion, and Kimmelstiel–Wilson nodules, along with arteriolar hyalinosis [7]. The histopathological features of DKD can be categorized into glomerular changes, tubulointerstitial changes, and histological classification [21]. DKD is characterized by glomerular and tubulointerstitial structural changes, including GBM thickening, mesangial expansion, glomerulosclerosis, vascular lesions, and tubular atrophy. Chronic hyperglycemia leads to GBM thickening, an early DKD marker, while podocytopathy contributes to albuminuria, glomerulosclerosis, and nephron loss [22]. Ectopic mesangial deposition of collagen I and III drives advanced DKD, while glomerular cell modifications and podocyte–endothelial interactions promote mesangial expansion via trophic and fibrogenic factors, offering potential therapeutic targets [23].

Elevated plasma glucose increases GLUT1/2-dependent basolateral glucose absorption, leading to tubular hypertrophy and epithelial remodeling in diabetes [24]. Early TBM thickening, driven by fibrogenic changes in proximal tubular cells, expands interstitial gaps in the extracellular matrix. Hypoxia further exacerbates structural and functional alterations, particularly in the proximal tubule [25]. Defective uptake and transcytosis are linked to hypoxia in diabetes and lead to microalbuminuria and the excretion of proteins such as gamma-glutamyl transpeptidase (γ-GT), neutrophil gelatinase-associated lipocalin (NGAL), and liver-type fatty acid-binding protein (L-FABP). The degree of tubular damage, interstitial fibrosis, and inflammation in the diabetic kidney is linked to the rise in these urine markers [26].

Over time, tubular epithelial cells undergo incremental and cumulative apoptosis in response to tubular enlargement, increasing hypoxia, and remodeling. Studies suggest that hypoxia plays a role in early kidney injury, while in later stages, advancing fibrosis amplifies hypoxia, further accelerating disease progression. In advanced DKD, up to 50% of glomeruli are linked to dilated, atrophic tubules, and 17% may be atubular, leading to nephron dropout and kidney function decline. GBM thickening, mesangial expansion, and podocyte changes contribute to hemodynamic stress, driven by hyperfiltration and inflammatory, trophic, and profibrogenic mediators [27].

DKD involves activated T cells and macrophage recruitment, driving disease progression. Albuminuria and GFR declines correlate with local and systemic inflammation. Nephron dropout and tubulointerstitial fibrosis, marked by increased matrix deposition, altered composition, and reduced turnover, contribute to kidney function loss. Interstitial changes may predict disease progression more strongly than glomerular alterations [28].

## 4. Evaluation

Full-blown DKD is a clinical syndrome characterized by overt proteinuria (urinary albumin creatinine ratio, UACR > 300 mg/g) and declining renal function. Screening for DKD is currently centered around assessing proteinuria and renal function. The natural history of classical diabetic nephropathy is characterized by progression in the stages from normo-albuminuria, microalbuminuria, to overt proteinuria, followed by a decline in renal function to the development of ESRD. The clinical diagnosis is based on proteinuria and renal function tests, while the confirmatory histological diagnosis in DN is through renal biopsy, showing characteristic pathological changes, such as mesangial expansion and glomerulosclerosis [21].

However, the trajectories of the renal function decline would vary from individual to individual depending on several factors, like patient characteristics, clinical parameters, and lifestyle factors. A subset of patients with diabetes may have renal function loss without significant proteinuria, which is known as non-proteinuric diabetic kidney disease [29].

Compared to proteinuric DKD, individuals with non-proteinuric DKD have fewer changes in the glomeruli, tubulointerstitium, and vasculature. Studies indicate that patients with this phenotype exhibit fewer glomerular lesions but more severe tubulointerstitial and vascular changes. Mechanisms contributing to tubular injury include oxidative stress and renal hypoxia, which damage tubular cells, as well as endoplasmic reticulum stress, leading to cell injury and apoptosis. Additionally, inflammation and fibrosis further drive renal function decline. These distinctions suggest that while glomerular damage dominates proteinuric DKD, non-proteinuric DKD progresses through tubular and interstitial pathways, emphasizing the need for tailored diagnostic and therapeutic approaches. Another novel variant of DKD is a subset of individuals with DM who present with non-proteinuric DKD. The prevalence varies from around 20% and 40%, in individuals with T1DM and T2DM, respectively [21,22,23].

The current standard screening tools like albuminuria and eGFR are insufficient in detecting early changes in the kidney. The biomarkers that reflect tubular injury and interstitial pathology are currently being explored in the evaluation of DKD. Biomarkers appear to be useful in the early identification of DKD and serve as complementary markers to albuminuria in the evaluation of DKD. Candidate biomarkers have been chosen based on in vitro and in vivo studies that indicate their potential significance in the pathophysiology of DKD [30]. These biomarkers are involved in inflammation, fibrosis, endothelial dysfunction, and tubular injury. Several serum and urine markers have been suggested as potential DKD indicators. The development of novel ‘omic’ techniques and the integration of multiple omics data, or multi-omics, hold great promise for the identification of DKD biomarkers [30].

The novel biomarkers that are being studied in DKD include (a) glomerular markers like transferrin, Type IV collagen, ceruloplasmin, and fibronectin; (b) tubular biomarkers like α1-microglobulin (A1M), Serum Cystatin C, L-FABP, N-acetyl-β-D-glycosaminidase (NAG), urinary immunoglobulin G and M, Kidney injury molecule-1 (KIM-1); and (c) markers of oxidative stress/inflammation like 8-hydroxy-2′-deoxyguanosine (8-OHdG), pentosidine, AGA, TNF-α, TNFAR-1/2, IL-6, and VEGF [31].

## 5. Glomerular Markers

Transferrin: A glomerular marker called transferrin has been shown to occur in urine even before development of micro-albuminuria. Urinary transferrin is thought to be a more sensitive indicator of glomerular injury in diabetic patients [32]. Increased urine transferrin excretion predicts the future development of microalbuminuria in type 2 diabetic individuals with normo-albuminuria, and urinary transferrin excretion and urinary albumin excretion in diabetic patients have a good linear association [33]. Few studies showed that type 1 diabetics who had normo-albuminuria had noticeably greater urine transferrin levels. Long-term studies on urinary transferrin will demonstrate the utility of this marker in DKD [34].

Type IV collagen: DKD is linked to specific urinary collagen fragments, indicating that alterations in the extracellular matrix and collagen turnover are markers of the molecular pathophysiology of diabetes. The reduction in eGFR is independently linked to significant urine excretion of type IV collagen [35]. In people with normo-albuminuria, a correlation between urine type IV collagen and eGFR reduction was also seen. Therefore, in the absence of increased albuminuria, measuring urine type IV collagen may assist in identifying individuals who are at risk of progressive deterioration in GFR [36]. More research is needed to determine how the findings of decreased collagen fragments found in proteomic investigations connect to the findings of increased full-length collagen urine excretion.

Ceruloplasmin: Diabetes patients with normo-albuminuria are known to have urinary ceruloplasmin, and a rise in this protein in the urine is predictive of the onset of microalbuminuria in these patients [33]. In vitro and animal studies have demonstrated that both AGEs and hyperglycemia raise ROS and other inflammatory mediators in the kidney, which may indicate that kidney-specific ceruloplasmin is elevated in DKD [37]. Histochemical staining of kidney biopsy specimens from DKD patients for ceruloplasmin revealed significant amounts of ceruloplasmin in the kidney, proving that kidney-specific ceruloplasmin was elevated in human DKD. Urine ceruloplasmin is a promising indicator of DKD; nevertheless, more research is required to fully understand its significance in comparison to albuminuria [38].

Fibronectin: Fibronectin is an example of extracellular-matrix (ECM) proteins that build up in the renal tubulointerstitium and mesangium during hyperglycemic stress. ECM protein deposition then triggers podocytes and renal tubular epithelial cells, which further develops to myofibroblasts and display fibroblast markers. Furthermore, renal fibroblasts multiply and aggravate DKD by generating glomerulosclerosis and fibrosis [39].

Immunoglobulins: Urinary IgG and IgM excretion is greater in diabetes and DKD patients. The development of glomerular diffuse lesions is correlated with urinary IgG excretion. IgG in the urine was a reliable indicator for predicting DKD onset. However, urinary IgM excretion has not been regarded as an early marker of DKD, since its excretion in urine is associated with severe injury of the glomerular capillary wall, while it is also a promising marker which may predict the eventual need for renal replacement therapy [33].

## 6. Tubular Biomarkers

α1-microglobulin (α1-MG): α1-MG can detect kidney tubular damage before significant glomerular dysfunction occurs. In the early stages of DKD, changes in tubular function often precede albuminuria, making α1-MG a valuable marker for identifying early kidney involvement even in normoalbuminuric diabetic patients [40]. Studies have shown that elevated urinary α1-MG levels correlate with disease severity and are predictive of faster progression to end-stage renal disease (ESRD) in diabetic patients [41]. As a marker, α1-MG could guide therapeutic interventions for slowing DKD progression.

Serum Cystatin C (CysC): CysC, a cysteine proteinase inhibitor, is a more accurate indicator of deteriorating GFR in DM patients than serum creatinine. Serum CysC showed a high diagnostic value for diabetic nephropathy with strong sensitivity and specificity in a meta-analysis by Liao X et al. [42]. It is freely filtered at the glomerular level due to its small size and positive charge, and the renal tubules thereafter fully reabsorb and metabolize it without secreting it. Consequently, serum CysC may indicate early alterations in renal function and a decline in eGFR, and it can be employed as a biomarker for the early diagnosis of AKI [43].

Neutrophil gelatinase-associated lipocalin (NGAL): NGAL is specifically generated by both neutrophils and damaged nephron epithelial cells. Renal damage in DKD is caused by oxidative stress and chronic inflammation, and high NGAL levels may help maintain the kidney’s inflammatory milieu [44]. NGAL levels correlated with declining renal function, highlighting its diagnostic utility for detecting early tubular damage. NGAL provides additional diagnostic information beyond albuminuria and the estimated glomerular filtration rate (eGFR). Patients with diabetes may benefit from early detection, treatment guidance, and outcome prediction made possible by monitoring their NGAL levels [45]. Higher NGAL levels in diabetic patients are associated with rapid progression to advanced stages of DKD and end-stage renal disease (ESRD). It is particularly valuable in cases where albuminuria levels do not fully reflect the underlying kidney damage, such as in normoalbuminuric DKD patients. Targeting pathways related to NGAL expression may help reduce tubular injury and oxidative damage [43].

Kidney injury molecule 1 (KIM-1): Patients with tubular injury tend to have higher serum levels of KIM-1, a type I transmembrane glycoprotein expressed in the proximal renal tubular cells’ apical membrane. Furthermore, plasma KIM-1 levels predicted an early decrease in eGFR and the progression of kidney disease, regardless of other variables like systolic blood pressure [30]. The soluble KIM-1 ectodomain is released into the urine when matrix metalloproteinases cleave KIM-1 from the cell surface. Chronic overexpression in tubular cells causes inflammation and interstitial fibrosis, whereas KIM-1 produced by acute tubular injury has anti-inflammatory effects through phagocytosis [46]. Urinary KIM-1 is, in fact, increasingly being shown to be a valuable biomarker of tubular damage. It is upregulated in response to tubular injury and mediates immune cell recruitment and cytokine release, both of which contribute to chronic inflammation and fibrosis in DKD [47]. Targeting KIM-1 expression or its downstream signaling pathways could help reduce inflammation and fibrosis in DKD. Preclinical studies indicate that suppressing KIM-1 activity or expression could limit tubular cell transformation into fibroblasts, which is a major driver of kidney fibrosis in DKD [48].

Fatty acid-binding protein 4 (FABP4): Renal proximal tubule cells contain FABP4, which is released in response to hypoxia brought on by a reduction in peritubular capillary blood flow. It is linked to abnormal lipid uptake in DKD, which is an important indicator of the disease’s onset and progression [49]. Increased expression of FABP4 is linked to ferroptosis and may change lipid deposition in DKD. Renal proximal tubule cells from DKD patients with iron accumulation in renal tubules and loss of mitochondrial cristae have increased expression of FABP4 [50]. However, it is unclear what mechanisms cause patients with DKD to have elevated FABP4 levels. FABP4 is widely expressed in endothelial cells, macrophages, and adipocytes and may serve as a diagnostic for the early identification of DKD [51]. The recent study highlights that while elevated levels of urinary NGAL and L-FABP are observed in DKD patients, these biomarkers are also increased in various non-diabetic CKD conditions, such as hypertensive nephrosclerosis and glomerulonephritis. This overlap suggests that neither NGAL nor L-FABP possesses sufficient specificity to serve as standalone indicators for DKD [52].

Several biomarkers have been identified for DKD, as explained above, each serving distinct roles in early detection, disease progression monitoring, and prognosis assessment. The specificity and sensitivity of these biomarkers [32], as outlined in Table 1, determine their reliability in clinical practice.

Reflecting the complex molecular changes connected to the course of DKD, urinary exosomes are essential sources of microRNAs (miRNAs) and show great promise as non-invasive diagnostics. Exosomal miRNAs are vital for cell-to-cell communication and have attracted more interest because of their stability and ability to show pathological changes in the kidney [53]. Research has found several exosome miRNAs showing changed expression patterns in people with diabetic nephropathy, suggesting possible diagnostic and prognostic value. While miR-15a-5p shows a decrease, specific miRNAs are elevated; miR-150-5p shows higher levels in patients with macroalbuminuria [54,55]. Differential expressions set a foundation for differentiating stages or severity levels in DKD. Increased levels of miR-342-3p and miR-30a-5p have been identified in type 2 diabetes patients with micro- or macroalbuminuria, highlighting their potential as early biomarkers for diagnosing diabetic nephropathy before the onset of overt clinical signs [56]. Moreover, several miRNAs show differential expressions in the urine exosomes of diabetic nephropathy patients: miR-21-5p, miR-23b-3p, and miR-15b-5p [57,58]. This increased the understanding of the molecular processes linked to diabetic nephropathy and the biomarker toolbox. The findings show that tracking exosomal miRNA signatures offers a dynamic and sensitive method for evaluating therapy response and disease progression, perhaps enhancing DKD’s clinical management.

Thus, biomarkers offer a more precise, predictive, and personalized approach to care and may contribute to better understanding of the underlying pathophysiology and to significantly improve patient outcomes.

## 7. Metabolomics

Metabolomics is a newer diagnostic modality being evaluated to assist in the early diagnosis and progression prediction of DKD. Certain metabolites, including lipid metabolites, branched-chain and aromatic amino acids, and certain fatty acids, are often altered in DKD patients, suggesting that they may serve as biomarkers [52]. Esterified and non-esterified fatty acids, carnitines, phospholipids, and metabolites involved in amino acid metabolism are among the biomarkers that have been determined by metabolomics. By linking lipid and amino acid metabolism to the development of the disease, metabolomics sheds light on the pathophysiological mechanisms of DKD [59]. By identifying patient-specific metabolic changes, metabolomics can inform individualized treatment plans while enhancing clinical results.

Numerous studies demonstrate that metabolomics, which uses urine and plasma analysis, is an efficient method for diagnosing DKD. Pena et al. found that a panel of biomarkers that combined baseline albuminuria and eGFR with plasma metabolites histidine and butenoylcarnitine and urinary hexose, glutamine, and tyrosine improved risk prediction of the transition from microalbuminuria to macroalbuminuria in DKD patients [60]. Numerous studies examined the potential for integrating metabolomics with proteomics or genomics/epigenomics to enhance the diagnosis of early DKD. Integrative serum proteomics and metabolomics data analysis, which aims to develop a model for predicting kidney damage based on a biomarker panel, could increase the accuracy of DKD prediction [61]. Alterations in proteomic biomarkers uromodulin and CD59 have been associated with DKD. The thick ascending limb of Henle (TAL) synthesizes a kidney-specific protein called uromodulin (umod). Umod reflects intact TAL cell mass and may serve as a biomarker for assessing tubular function. It is produced in response to inflammation and interstitial fibrosis. Studies have shown that changes in uromodulin concentrations correlate with kidney function decline, suggesting its utility as a biomarker for early DKD detection and progression monitoring [62]. CD59 is a complement regulatory protein which has a protective role against complement-mediated damage primarily in the glomerular cells. It inhibits membrane attack complex (MAC). CD59 is produced by the mesangial and interstitial cells of the kidney. Hyperglycemia causes glycation of CD59 (glycated CD59), which leads to complement-mediated glomerular damage [63]. However, its specificity and effectiveness as a biomarker for DKD are still being investigated. Further research is required to clarify the role of CD59 in DKD diagnosis and prognosis.

Furthermore, mitochondrial dysfunction in diabetes plays a significant role in the development of complications of diabetes, especially DKD, because of a change in specific metabolites during disease progression. Measuring mitochondrial function may help in understanding the pathogenesis of DKD and in targeting the therapeutic approach for DKD management [64]. Characterization of metabolome changes in response to various treatment approaches is necessary. Furthermore, to strengthen the results, more research is required with larger cohorts and longer follow-up periods, ideally involving patients in the early stages of the disease [61]. As DKD progresses, the metabolome alters, and distinct patterns of clinical advancement must be identified [65].

## 8. Genetic Markers

Both genome-wide association studies (GWAS) and exome-wide association studies (EWAS) have discovered an increasing number of genetic loci for DKD during the last five years. Until now, about 80 genetic loci for albuminuria, DKD, or eGFR in diabetes have achieved genome-wide statistical significance [66]. Several genetic loci that have been found currently, which has made it possible to compare the results and determine the genetic overlap between DKD in T1DM and T2DM and CKD in the general population. People with diabetes, particularly those with T2DM, appear to have eGFR influenced by the general population loci for eGFR [67]. Comparing the results and the genetic overlap between DKD and CKD in the general population in T1DM and T2DM is now possible due to the number of discovered genetic loci. It appears that the eGFR loci found in the general population also influence eGFR in people with diabetes, particularly those with T2DM [68].

The progression of DKD can be predicted with the help of DNA methylation indicators. However, recent research has concentrated on the later phases of kidney disease, when renal failure has set in, or the albumin excretion rate is significantly elevated. Despite not yet being used as widely in risk prediction as polygenic risk scores, DNA methylation scores exhibit considerable promise because they consider data from both the environment and genes. Methylation scores outperformed polygenetic risk scores in a recent study, improving the prediction of a variety of clinical diagnoses and features, including renal disease [69]. However, there are restrictions on causation, the duration of impact, and the target tissue due to the dynamic nature of methylation and its tissue specificity. Causality can be addressed by including genetic information, and new single-cell sequencing technologies may make it easier to conduct more focused analyses in the future. For example, they may allow researchers to examine the causal consequences of DNA methylation in the kidneys at the single-cell level [68].

The most researched epigenetic alteration is DNA methylation, which takes place at the cytosine bases of the DNA sequences cytosine–phosphate–guanine dinucleotide sites (CpGs). There are other epigenetic changes beyond DNA methylation, like histone modifications (acetylation and methylation), and their connection to DKD has also been investigated. For instance, DKD has been linked to the dysregulation of histone H3 lysine 27 trimethylation (H3K27me3) in TGF-b1-induced gene expression [70].

The potential biomarkers in DKD diagnostics are shown in Figure 1 below:

## 9. Current Disease-Modifying Approaches for DKD

Prompt glycemic control with appropriate lifestyle changes, including regular structured physical activities and a balanced diet to prevent hyperglycemia, are the corner stones in the therapeutic approach to patients with diabetes. As smoking is an important vascular risk factor, smoking cessation strategies are also highly important to prevent and avoid the progression of diabetes related micro-vasculopathy leading to DKD.

The management of DKD has advanced alongside our deepening understanding of its complex pathophysiological mechanisms, which involve hemodynamic, metabolic, and inflammatory pathways. These interconnected pathways are crucial in the onset and progression of DKD. Multifaceted intervention is needed to target arresting kidney disease progression and modifying CVD risk factors in patients with DKD. Lifestyle modification, including dietary counseling, encouraging physical activity and smoking cessation, along with management of hypertension, dyslipidemia and hyperglycemia, and the use of drugs with renoprotection properties have shown a reduction in renal as well as cardiovascular events [71,72]

Targeting abnormalities of RAAS has been the primary and well-established therapy for DKD. Angiotensin Convertase Enzyme (ACE) inhibitors and Angiotensin receptor blockers (ARBs) have demonstrated a reduction in proteinuria and progression of DKD in well-established clinical trials of diabetic nephropathy [73,74,75,76,77]. However, a combination of these drugs was not extra-beneficial and was associated with higher adverse effects [78,79]. They are indicated in all diabetic patients with hypertension and/or proteinuria [80].

For individuals with DKD, RAAS inhibitors (ACE-I or ARBs) are the recommended antihypertensive medications. Captopril treatment showed significant improvement in people with DKD from T1DM but no improvement in Type 2 diabetic nephropathy. In individuals with type 2 diabetes and nephropathy, the ARB losartan had positive benefits on cardiovascular and renal outcomes [74]. Losartan reduced mortality, ESRD, and the chance of tripling the risk of serum creatinine levels. When it came to nephropathy caused by T2DM, irbesartan and losartan both had comparable reno-protective effects. Remarkably, this defense does not depend on blood pressure regulation. Furthermore, even after the conventional ARB administration was put into place, DKD continued to progress [71].

Mineralocorticoid receptor antagonists (MRA) suppress the action of aldosterone; the product of RAAS activation exhibits antihypertensive effect and reduces proteinuria [81]. Apart from the distal nephron, fibroblasts, macrophages, podocytes, and vascular cells are among the cell types that express mineralocorticoid receptors (MRs). RAAS activation brought on by a decrease in circulating-plasma volume encourages the release of aldosterone [82]. Following its contribution to MR activation, aldosterone causes potassium excretion and sodium reabsorption. High sodium consumption activates MRs, which causes hypertension and glomerular injury and fibrosis [82]. Obesity, insulin resistance, hyperglycemia, and dyslipidemia all increase MR expression, which in turn raises profibrotic and inflammatory factors (extracellular matrix proteins, PAI-1, TGF-β, and CTGF), ultimately leading to the development of DKD [83]. Therefore, MRA has emerged as one of the cornerstones in the disease-modifying approaches against DKD in recent years. Spironolactone, the first drug molecule of this class has a steroidal molecular structure and has several undesirable side effects such as gynecomastia, menstrual irregularities, and breast tenderness due to its androgenic properties, especially at higher doses.

Non-steroidal mineralocorticoid receptor antagonists (NS-MRAs), such as finerenone, apararenone, esaxerenone, and ocedurenone, are being studied in DKD. These vary significantly from steroidal MRAs in that they diffuse between kidney and heart tissues instead of acting on the kidney alone [84]. In the Finerenone in the Reducing Kidney Failure and Disease Progression in Diabetic Kidney Disease (FIDELIO-DKD) and Finerenone in Reducing cardiovascular mortality and morbidity in Diabetic Kidney Disease (FIGCARO-DKD) clinical trials, participants with T2DM and DKD treated with ARB or ACEI as the standard care plus finerenone showed a significant decrease in albuminuria, blood pressure, and the risks of atherosclerotic disease and heart failure (FIDELIO-DKD) research [85]. Since there is no definitive outcome evidence, other NS-MRAs are largely recommended for blood pressure control, while finerenone is the only one licensed for protection against cardiorenal events. It has been noted that esaxerenone decreases the excretion of albumin. However, esaxerenone’s long-term cardiovascular and renal advantages have not yet been determined, and its regulatory approval is still relatively new [86].

MRA group of drugs like spironolactone and eplerenone demonstrated beneficial effects in decreasing albuminuria, inflammation, and fibrosis among patients with DKD [87]. However, increased risk of hyperkalemia limits their use in DKD patients with impaired renal function. The newer non-steroidal MRAs (NS-MRAs) like finerenone and esaxerenone with greater selectivity for mineralocorticoid receptor appears promising with fewer complications in treating DKD. The non-steroidal MRAs have demonstrated a significant reduction in proteinuria, suggesting beneficial effects in DKD. Finerenone has demonstrated significant improvement in renal outcomes as well as a reduction in adverse cardiovascular events in phase III clinical trials [88,89]. The concerns for hyperkalemia and acute kidney injury with these drugs suggest monitoring of serum potassium and renal function. Despite the longstanding importance of RAAS blockade as a foundational treatment, significant residual risk persists.

Sodium–Glucose Cotransporters 2 Inhibitors (SGLT-2Is): SGLT-2Is have been demonstrated to lower the risk of ESRD, or kidney disease-related death, in people with T2DM. Several mechanisms have been proposed for the nephroprotective effects of SGLT-2Is, which include reduction in glomerular hyperfiltration by increasing the sodium reaching the macula dense cells of the distal tubules, thereby reversing the dilation of afferent arterioles and glomerular hyperfiltration by the tubule–glomerular feedback (TGF), reduction in renal hypoxia by reducing Na-K ATPase activity and improving renal cortex oxygenation, induction of mild ketosis, augmenting cellular bioenergetics, and reduction in oxidative stress [90,91]. These mechanisms may reduce the generation of ROS, inflammatory markers, fibrotic factors, ketone production, and AGE development in proximal tubular cells [82]. The renal outcome trials as well as post hoc analysis of cardiovascular outcomes trials (CVOTs) of SGLT-2Is (Empagliflozin, Dapagliflozin, and Canagliflozin) have demonstrated significant reduction in adverse renal outcomes like doubling of serum creatinine, decline in renal function and development of ESRD or the need for renal replacement therapy [90,92,93,94,95]. The identification of the urine proteome impacted by SGLT2i opened new possible target sites and pathways, particularly those associated with inflammation and wound healing.

Glucagon-like peptide-1 receptor agonists (GLP-1RAs): Incretin-based therapy with GLP-1RAs have also shown significant benefits in managing patients with DKD. These drugs have shown significant reduction in proteinuria and rate of decline of renal function irrespective of the glycemic status. The reno-protective roles of GLP-1RAs are mediated through ameliorating oxidative stress, cellular apoptosis, and fibrosis. Inhibition of NF-κB signaling by GLP-1RAs effectively suppress the expression of proinflammatory cytokines like TNF-α, IL-1, IL-6, chemokines, and fibrotic factors like TGF-β [86]. Despite the best possible treatment with metformin and SGLT2 inhibitors, GLP-1RAs have been approved for the treatment of hyperglycemia, prevention of atherosclerotic CV disease, and/or treatment of DKD patients at high risk for CV events [96]. Furthermore, GLP-1RAs aid in weight loss, which increases its adaptability in the treatment of diabetes. Independent of glycemic control, a secondary analysis of glycemic lowering and CV outcome trails has validated the reno-protective effects of GLP-1RA in T2DM by lowering albuminuria and delaying the deterioration in eGFR [97]. By reducing oxidative stress, cellular apoptosis, and fibrosis, GLP-1RAs have been shown to have reno-protective effects. GLP-1RAs can reduce ROS production and inhibit NF-κB activation, which further lowers the expression of fibrotic factor and cytokines production [96]. Semaglutide treatment decreased the probability of long-term decrease in eGFR by 40% and may lead to lowered rates of new or worsening nephropathy. Semaglutide treatment decreased the probability of long-term reductions in eGFR by 40% and may lead to decreased rates of new or worsening nephropathy [98]. GLP-1RAs and dipeptidyl peptidase 4 (DDP4) inhibitors can reduce blood pressure and body weight in addition to decreasing glucose [99]. Elevated leptin production in obese people causes proinflammatory and chemokine production, which ultimately leads to podocyte loss, fibrosis, and renal failure. DDP4 inhibitors have not been able to prevent the reduction in eGFR and have only modestly improved albuminuria [96]. However, recent drug discoveries have brought about a major paradigm shift in treatment strategies, advocating for a multi-pillared approach to optimize outcomes [17,97].

## 10. Clinical Studies on Emerging Therapies in DKD

Large-scale clinical trials provide most of the current evidence supporting medical treatments for diabetic kidney disease (DKD). Numerous categories of drugs have been introduced and assessed in extensive randomized clinical studies to decrease the burden of decreasing renal function in DM patients [100].

CREDENCE, DAPA-CKD, and EMPA-KIDNEY are the three main landmark clinical trials for SGLT2i [101,102,103]. Two large randomized clinical trials (FIDELIO-DKD and FIGARO-DKD) [104,105] and their pooled analysis (FIDELITY) have demonstrated the cardiorenal effects of finerenone [85]. GLP-1RA was most recently presented as a possible DKD treatment option in the FLOW trial [106]. Adults with type 2 diabetes were included in these clinical studies either as a subgroup factor or as the primary inclusion criterion. Except for the sub-group in EMPA-KIDNEY, all the DKD clinical trials excluded patients with T1DM. Renal death, end-stage kidney disease (ESKD) requiring ongoing dialysis or transplantation, a sustained eGFR < 15 mL/min/1.73 m^2^ (except EMPA-KIDNEY), or a sustained fall in eGFR from baseline were the criteria used to characterize kidney outcomes in all DKD clinical trials. However, the standards for a sustained drop in eGFR varied substantially throughout these trials [104].

DKD progression is still an issue, despite recent advancements in treatment alternatives, such as SGLT2i, MRA, and/or GLP-1RA in combination with RAAS inhibitors, showing encouraging results in clinical trials [100]. Finerenone showed benefit; however, approximately 7% of FIDELITY participants were receiving either SGLT2i or GLP-1RA [107]. Although 16% of patients received SGLT2i at baseline, there was no discernible variability in the semaglutide effect in FLOW; however, the statistical power of effect modification analysis remained limited. Nonetheless, finerenone was not authorized for use in the FLOW and SGLT2i trials. By comparing the renal, cardiovascular, and mortality benefits of combination therapy (SGLT2i + NS-MRA and GLP-1RA) to conventional care alone, the evidence from the pooled data from clinical studies showed the potential benefits of combination therapy. According to the study, combination therapy may improve the major adverse cardiovascular-event-free survival and slow the progress of CKD [108]. The CONFIDENCE clinical trial is presently underway and aims to examine the impact of the combination of finerenone and SGLT2i. The results of this trial and empirical data will help shed further information on the efficacy and affordability of various treatments [100].

## 11. Future Therapeutic Targets and Drug Development

Several new agents have significantly advanced the management of DKD, offering clear protective benefits not only in slowing down kidney disease progression but also in reducing cardiovascular risk. Despite the introduction of ARBs, ACE inhibitors, NS-MRAs and SGLT-2Is, current therapies still fall short of fully halting DKD progression. GLP-1RAs, a promising class of antidiabetic agents, show potential as renal protectors, effectively slowing DKD progression [109].

Additionally, other agents like pentoxifylline (PTF), selonsertib, and baricitinib, with their anti-inflammatory and antifibrotic properties are being studied as potential future DKD therapies [110]. NF-E2–related factor 2 (Nrf2) activators with protective effects against oxidative stress and chronic inflammation are a novel class of drugs with the proposed potential to improve the GFR of DKD patients, which are being explored for treating DKD [111]. Hypoxia-inducible factor prolyl hydroxylase inhibitors have shown benefits in CKD in preliminary experimental studies. The renal protective effects are mediated by mainly reducing renal hypoxia resulting in a reduction in the tubulointerstitial damage and progression of CKD [112,113].

The epigenetic alterations, like aberrant DNA methylation and histone modifications in response to hyperglycemia or hypoxia, are stored as cellular memory and can progressively lead to irreversible renal damage [114]. Drugs targeting these epigenetic alterations are being explored for the treatment of DKD. Histone deacetylase inhibitors, such as vorinostat, valproate, sodium butyrate, and trichostatin, have shown promising results in DKD rat models, where they have been reported to reduce proteinuria and improve oxidative stress, fibrosis, glomerular damage, and inflammation [115]. The ability to fully elucidate these epigenetic changes in DKD could pave the way for the development of innovative drugs aimed at reversing disease progression. Recent advancements in gene therapy have identified potential genetic targets for DKD, offering novel opportunities to modify disease progression through gene-based interventions. Additionally, aberrant histone methylation contributes to DKD pathogenesis, and targeting these epigenetic alterations with specific inhibitors presents a potential therapeutic approach [68]. There is also considerable interest in microRNA (miRNA) and long non-coding RNA (lncRNA) modulation by antagonists in the progression of DKD. Similarly, there is significant attention on mesenchymal stem cells (MSCs). MSCs have been shown to reduce inflammation and fibrosis in kidneys as well as in other diabetic vascular complications [116].

The development of novel therapeutics targeting inflammation, fibrosis, oxidative stress, hypoxia, and epigenetic modifications offers promising avenues for future DKD management. These therapeutic targets have been the new hope for DKD therapy. Additionally, large-scale clinical trials with rational designs are required for invalidation. The current and emerging therapeutic strategies based on the pathobiology of DKD are shown in Figure 2 below:

## 12. Conclusions

A multidisciplinary approach, combining lifestyle modifications with drug therapy, can effectively decelerate DKD progression. Traditional biomarkers, such as albuminuria and eGFR, though widely used, have limitations in early diagnosis and progression monitoring. Novel biomarkers, including circulating microRNAs, inflammatory markers, and metabolites, offer promising insights into the underlying molecular pathways of DKD and hold potential for improved diagnosis, prognosis, and personalized treatment strategies. Emerging therapies are redefining the management landscape of DKD by targeting novel pathways such as inflammation, oxidative stress, and fibrosis. Diabetes management should include prompt lifestyle changes along with smoking cessation and rational use of newer antidiabetic medications with vascular-disease-modifying properties. SGLT-2Is and GLP-1RAs have demonstrated significant renal protection, marking a paradigm shift in treatment recently. Furthermore, ongoing clinical trials on novel agents aim to evaluate their potential effectiveness in the management of DKD. The expansion of therapeutic options marks a new era in DKD management, offering hope for more impactful patient care. Furthermore, new insights into metabolic memory mechanisms, driven by AGEs and epigenetic changes in the kidney, suggest that inhibitors targeting AGE and histone modifications could become groundbreaking treatments for DKD.

## Figures and Tables

**Figure 1 diagnostics-15-00973-f001:**
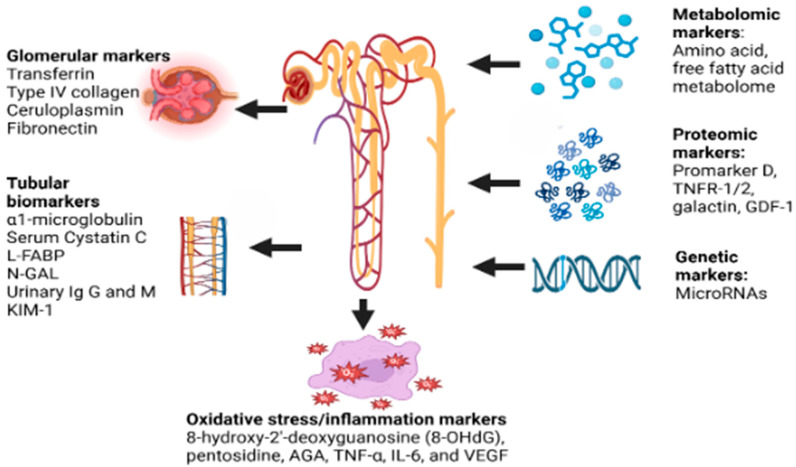
Potential biomarkers in DKD. FABP—liver fatty acid-binding protein, N-GAL—neutrophil gelatinase-associated lipocalin, KIM-1—kidney injury molecule 1, Ig G and M—immunoglobulin G and M, RNA—ribonucleic acid, TNFR-1/2—tumor necrosis factor receptor 1/2, GDF-1—growth differentiation factor 1, AGE—advanced glycation end-products, TNF-α—tumor necrosis factor alpha, IL-6—interleukin-6, VEGF—vascular endothelial growth factor.

**Figure 2 diagnostics-15-00973-f002:**
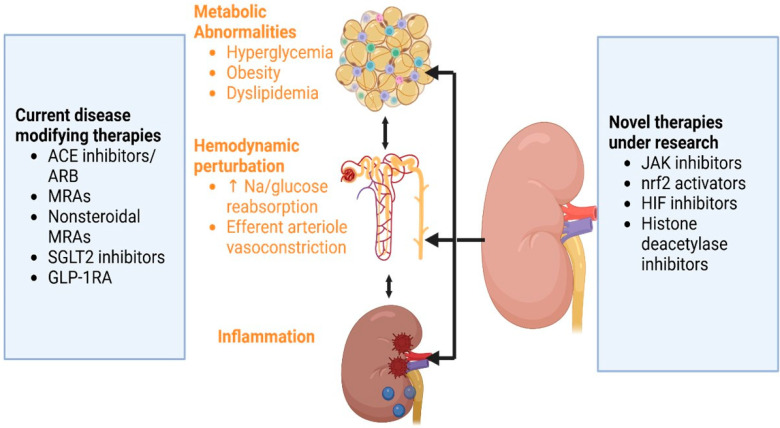
Pathophysiology of diabetic nephropathy and the therapeutic strategies. ACE: angiotensin convertase enzyme; ARB: angiotensin II blocker; MRAs: mineralocorticoid receptor antagonists; GLP-1 RA: glucagon-like insulinotropic peptide-1 receptor agonist; Na: sodium; JAK: Janus kinase; HIF: hypoxia-inducible factor; Nrf2: nuclear factor erythroid 2-related factor 2; SGLT2: Sodium-glucose cotransporter-2.

**Table 1 diagnostics-15-00973-t001:** Sensitivity and specificity of biomarkers in DKD [32].

Biomarker	Sensitivity (%)	Specificity (%)	Role in DKD
Serum NGAL	79	61	Early biomarker of DKD, reflecting tubular injury and inflammation
Urine NGAL	80	61	Early tubular injury marker, elevated before albuminuria
Serum Cystatin C	68	90	Estimation of kidney function
Urine L-FABP	98	90	Reflects tubular damage and oxidative stress
Urine NGAL/Creatinine	60	87	Early tubular injury in DKD, aiding in the detection of kidney damage
Serum Uromodulin	95	43	Tubular health indicator, inversely related to DKD progression
Urine KIM-1	79	51	Early tubular injury marker, associated with DKD progression
Urine Transferrin	47	98	Early glomerular injury marker, detected before albuminuria
Urine IgG	74	93	Glomerular permeability dysfunction in DKD

Note: NGAL—neutrophil gelatinase-associated lipocalin, L-FABP—liver-type fatty acid-binding protein, KIM-1—Kidney injury molecule-1.

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
