# Peer review of "Emerging Biomarkers and Innovative Therapeutic Strategies in Diabetic Kidney Disease: A Pathway to Precision Medicine"

_diagnostics, 2025, doi:10.3390/diagnostics15080973_

Round 1
Reviewer 1 Report
Comments and Suggestions for Authors
Comments and questions
I. Grammatical & Orthographical Errors
Formatting/Consistency:
- Line 39: “not yet fully elusive” → Replace “elusive” with “elucidated” (elusive = hard to find; elucidated = explained).
- Line 42: “newer understandings” → Replace with “recent advances” for clarity.
- Line 90: “APOL-l” → Typo; correct to “APOL1”.
- Line 91: “nondiabetic CKD” → Hyphenate as “non-diabetic”.
- Line 137: “gamma-glutamyl transpeptidase (γGT)” → Use consistent Greek letter formatting: “γ-GT”.
- Line 169: “Nonproteinuric” → Hyphenate as “non-proteinuric”.
Syntax/Flow:
- Line 38: “the pathobiological mechanisms for DN are not yet fully elusive” → Awkward phrasing; revise to “the pathobiological mechanisms of DN remain incompletely understood”.
- Line 54: “mi-crovascular” → Split incorrectly; correct to “microvascular”.
- Line 86: “T1D” vs. “T1DM” → Use consistent terminology (e.g., T1DM throughout).
- Line 106: “Pathogenesis of DN is complex” → Add “The” at the beginning: “The pathogenesis…”.
II. Factual Accuracy & Content Remarks
- Genetic Factors (Lines 82-98):
- The protective role of COL4A3 variants (rs55703767) in T1DM is oversimplified. Recent studies (Salem, R.M.; Todd, J.N.; Sandholm, N.; Cole, J.B.; Chen, W.M.; Andrews, D.; Pezzolesi, M.G.; McKeigue, P.M.; Hiraki, L.T.; Qiu, C.; et al. Genome-Wide Association Study of Diabetic Kidney Disease Highlights Biology Involved in Glomerular Basement Membrane Collagen. J. Am. Soc. Nephrol. 2019, 30, 2000–2016) show these variants reduce albuminuria risk but do not universally prevent collagen abnormalities. Clarify this.
- APOL1 G1/G2 alleles are linked to non-diabetic CKD, but emerging evidence (2024) suggests they may also accelerate DKD progression in Africans. Update this section. (Pollak MR, Friedman DJ. APOL1 and APOL1-Associated Kidney Disease: A Common Disease, an Unusual Disease Gene - Proceedings of the Henry Shavelle Professorship. Glomerular Dis. 2023 Jan 25;3(1):75-87. doi: 10.1159/000529227. PMID: 37113494; PMCID: PMC10126737).
- Epigenetics (Lines 95-103):
- The statement that “intensively managing hyperglycemia barely lowers DKD risk” is misleading. Cite DCCT/EDIC trials showing glycemic control reduces early DKD but not late-stage disease.
- Biomarkers (Lines 170-180):
- Urinary NGAL and L-FABP are mentioned, but recent review (Chen Y, Liu X, Shengbu M, Shi Q, Jiaqiu S, Lai X. Biomarkers: New Advances in Diabetic Nephropathy. Natural Product Communications. 2025;20(2). doi:10.1177/1934578X251321758) question their specificity for DKD vs. other CKD etiologies. Acknowledge this limitation.
- Missing discussion of proteomic biomarkers (e.g., CD59, uromodulin) validated in several studies.
III. Critical Questions & Literature-Based Remarks
- Non-Proteinuric DKD (Lines 168-169):
- The manuscript states that non-proteinuric DKD exists but does not address its prevalence or mechanisms. How do tubular vs. glomerular injury pathways differ here?
- Hypoxia & Tubular Injury (Lines 133-135):
- The role of hypoxia is presented as definitive, but some studies argue hypoxia is a consequence of fibrosis, not a driver. Can you please clarify ?
- JAK/STAT Pathway (Line 69):
- JAK inhibitors (e.g., baricitinib) show promise in DKD trials (Tuttle KR, Brosius FC 3rd, Adler SG, Kretzler M, Mehta RL, Tumlin JA, Tanaka Y, Haneda M, Liu J, Silk ME, Cardillo TE, Duffin KL, Haas JV, Macias WL, Nunes FP, Janes JM. JAK1/JAK2 inhibition by baricitinib in diabetic kidney disease: results from a Phase 2 randomized controlled clinical trial. Nephrol Dial Transplant. 2018 Nov 1;33(11):1950-1959. doi: 10.1093/ndt/gfx377. PMID: 29481660; PMCID: PMC6212720.). Should these therapies be highlighted alongside TGF-β antagonists?
- Clinical Translation:
- How do novel biomarkers (e.g., urinary exosomes) translate to clinical practice? Discuss cost, accessibility, and validation in diverse populations.
IV. Overall Recommendations
- Differentiate DKD (clinical diagnosis) vs. DN (histological diagnosis) consistently.
- Consider a table comparing biomarker sensitivity/specificity.
This manuscript provides a solid foundation but requires updates to reflect recent advancements and clearer articulation of contentious topics. With revisions, it will be adequate for publication in Diagnostics.
Comments on the Quality of English LanguageThere are some orthographical and grammatical that i have mentioned and that they need correction. Moreover, there are some phrases that need to be corrected.
Author Response
REVIEWER 1 |
||
A. Grammatical & Orthographical Errors – We have made every effort to modify with revision with necessary changes to improve the language to make the text clearer. |
||
Formatting/Consistency
|
||
Line 39 |
“not yet fully elusive” → Replace “elusive” with “elucidated” (elusive = hard to find; elucidated = explained). |
Changes have been made as per the suggestion in line 39. |
Line 42 |
“newer understandings” → Replace with “recent advances” for clarity |
Replaced with “recent advances” in line 42 |
Line 90 |
“APOL-l” → Typo; correct to “APOL1”. |
“APOL-l” correct to “APOL1” in line 75 |
Line 91 |
“nondiabetic CKD” → Hyphenate as “non-diabetic”. |
Corrected as per suggestion |
Line 137 |
“gamma-glutamyl transpeptidase (γGT)” → Use consistent Greek letter formatting: “γ-GT”. |
Corrected as per suggestion |
Line 169 |
“Nonproteinuric” → Hyphenate as “non-proteinuric”. |
Corrected as per suggestion |
Syntax/Flow |
||
Line 38 |
“the pathobiological mechanisms for DN are not yet fully elusive” → Awkward phrasing; revise to “the pathobiological mechanisms of DN remain incompletely understood”. |
The sentence is rephrased and mentioned in line 38 |
Line 54 |
“mi-crovascular” → Split incorrectly; correct to “microvascular” |
Corrected as per suggestion |
Line 86 |
“T1D” vs. “T1DM” → Use consistent terminology (e.g., T1DM throughout). |
Consistent terminology is used |
Line 106 |
“Pathogenesis of DN is complex” → Add “The” at the beginning: “The pathogenesis…”. |
Corrected as per suggestion in line 93. |
B. Factual Accuracy & Content Remarks |
||
Genetic Factors: (Lines 82-98) |
The protective role of COL4A3 variants (rs55703767) in T1DM is oversimplified. Recent studies (Salem, R.M.; Todd, J.N.; Sandholm, N.; Cole, J.B.; Chen, W.M.; Andrews, D.; Pezzolesi, M.G.; McKeigue, P.M.; Hiraki, L.T.; Qiu, C.; et al. Genome-Wide Association Study of Diabetic Kidney Disease Highlights Biology Involved in Glomerular Basement Membrane Collagen. J. Am. Soc. Nephrol. 2019, 30, 2000–2016) show these variants reduce albuminuria risk but do not universally prevent collagen abnormalities. Clarify this. |
This has been clarified, and the sentence has been rephrased in line 69. |
|
APOL1 G1/G2 alleles are linked to non-diabetic CKD, but emerging evidence (2024) suggests they may also accelerate DKD progression in Africans. Update this section. (Pollak MR, Friedman DJ. APOL1 and APOL1-Associated Kidney Disease: A Common Disease, an Unusual Disease Gene - Proceedings of the Henry Shavelle Professorship. Glomerular Dis. 2023 Jan 25;3(1):75-87. doi: 10.1159/000529227. PMID: 37113494; PMCID: PMC10126737). |
This has been clarified, and the sentence has been rephrased in line 75. |
Epigenetics (Lines 95-103) |
The statement that “intensively managing hyperglycemia barely lowers DKD risk” is misleading. Cite DCCT/EDIC trials showing glycemic control reduces early DKD but not late-stage disease. |
The statement has been rephrased and the DCCT/EDIC trials have been cited and mentioned in lines 84–88. |
Biomarkers (Lines 170-180) |
Urinary NGAL and L-FABP are mentioned, but recent review (Chen Y, Liu X, Shengbu M, Shi Q, Jiaqiu S, Lai X. Biomarkers: New Advances in Diabetic Nephropathy. Natural Product Communications. 2025;20(2). doi:10.1177/1934578X251321758) question their specificity for DKD vs. other CKD etiologies. Acknowledge this limitation. |
The limitation has been acknowledged and mentioned in lines 265-269 |
|
Missing discussion of proteomic biomarkers (e.g., CD59, uromodulin) validated in several studies |
The details of the same have been explained in lines 329-341 |
C. Critical Questions & Literature-Based Remarks
|
||
Non-Proteinuric DKD (Lines 168-169) |
The manuscript states that non-proteinuric DKD exists but does not address its prevalence or mechanisms. How do tubular vs. glomerular injury pathways differ here? |
This has been clarified, and explained in lines 148-159. |
Hypoxia & Tubular Injury (Lines 133-135) |
The role of hypoxia is presented as definitive, but some studies argue hypoxia is a consequence of fibrosis, not a driver. Can you please clarify? |
This has been clarified, and mentioned in line 125. |
JAK/STAT Pathway (Line 69) |
JAK inhibitors (e.g., baricitinib) show promise in DKD trials (Tuttle KR, Brosius FC 3rd, Adler SG, Kretzler M, Mehta RL, Tumlin JA, Tanaka Y, Haneda M, Liu J, Silk ME, Cardillo TE, Duffin KL, Haas JV, Macias WL, Nunes FP, Janes JM. JAK1/JAK2 inhibition by baricitinib in diabetic kidney disease: results from a Phase 2 randomized controlled clinical trial. Nephrol Dial Transplant. 2018 Nov 1;33(11):1950-1959. doi: 10.1093/ndt/gfx377. PMID: 29481660; PMCID: PMC6212720.). Should these therapies be highlighted alongside TGF-β antagonists? |
Although baricitinib may attenuate the pro-fibrotic response of TGF-β, there are other anti-fibrotic mechanisms of action. Therefore, we have not be highlighted as a TGF-β antagonist.
|
Clinical Translation |
How do novel biomarkers (e.g., urinary exosomes) translate to clinical practice? Discuss cost, accessibility, and validation in diverse populations. |
As suggested, this has been explained in lines 282-299 |
D. Overall Recommendations |
||
|
Differentiate DKD (clinical diagnosis) vs. DN (histological diagnosis) consistently. |
The clinical and histological diagnosis of Diabetic nephropathy has been added ( lines 139 to 142) |
|
Consider a table comparing biomarker sensitivity/specificity. |
Table 1 has been added for the same. |
Reviewer 2 Report
Comments and Suggestions for Authors
This manuscript is very informative. I suggest some revision.
1- the manuscript is too long. I suggest that section 2 and 3 about pathobiology and structural changes delete of at least summarize.
2- line 33 dreaded change to dreadful
3- line 58 change to Glomerular Basement Membrane (GBM)
4- numbering after section 8 need to correction and after correction number 8 about genetic markers replace before section 11 or 12 future of therapeutic targets and drug development.
5- the last section about future of therapeutic targets and drug development is less informative I suggest that this section more explain.
Author Response
REVIEWER 2 |
||
|
The manuscript is too long. I suggest that section 2 and 3 about pathobiology and structural changes delete of at least summarize. |
Section 2 and 3 about pathobiology and structural changes has been concised. |
line 33 |
dreaded change to dreadful. |
Corrected as per suggestion |
line 58 |
change to Glomerular Basement Membrane (GBM) |
Corrected as per suggestion |
|
Numbering after section 8 need to correction and after correction number 8 about genetic markers replace before section 11 or 12 future of therapeutic targets and drug development. |
We would like to thank the reviewer for pointing this out. However, we would like to point out that we have mentioned DNA methylation and histone modifications again in part 11 of our manuscript. We are of the humble opinion that the order of this manuscript should not be changed because the flow in its present form of diagnostics followed by therapeutics is maintained. |
|
The last section about future of therapeutic targets and drug development is less informative I suggest that this section more explain. |
We thank the reviewer for pointing out the brevity of this section. We have provided additional information regarding the future of DKD therapy and have covered most of the newer therapeutics mentioned in literature. |
Reviewer 3 Report
Comments and Suggestions for Authors
To the Editor of Diagnostics MDPI
It was with interest that I read ““Emerging Biomarkers and Innovative Therapeutic Strategies in 2 Diabetic Kidney Disease: A Pathway to Precision Medicine” manuscript by Shetty S et al. The Authors update the current and future diagnostic and therapeutic landscapes against diabetic nephropathy (DKD). They examine in depth the role of emerging novel biomarkers with the aim to show their potential role to improve diagnosis, prognosis and personalized treatment strategies
The review is well written and interesting, according to the purpose of this Journal. We thank the Authors for the effort put into this narrative review, it was deeply appreciated.
Here some suggestions:
- Line 45. Pathobiology:
a brief comment on the importance of kidney biopsy in diagnosing exactly the diabetic kidney disease would be done.
- Line 222 paragraph 6. tubulary biomarkers:
Tubular biomarkers, it is not clear why immunoglobulins belong to this category and not to glomerular markers.
the authors conclude that they may offer a more precise approach to care: given the expectations related to metabolomics, why are they not used in clinical practice given the impressive clinical, social and even economic impact of DKD? a comment from the authors?
- Line 317
“genome-wide association studies (EWAS)” should be maybe “esome-wide association studies”
- Line 365 9. Current disease modifying approaches
the authors, in addition to the promising results offered by the new hypoglycemic drugs, in discussion section and not only, briefly in conclusion, cannot help but mention the importance of lifestyle, strict dietary control in order to reduce hyperglycemia and physical activity, in addition to quitting smoking
- Line 382 MRA:
The paragraph could be further elaborated, better explaining the role of the drugs in light of the physiological regulatory mechanism. In addition, examples of drugs in this category could be included.
- Line 464, 10.paragraph. Clinical studies
It might be useful to add a table summarizing what is discussed in paragraph 10, summarizing the characteristics of clinical trials on emerging therapies in DKD, this would lighten the text a bit.
- Please, add references when you cites articles or studies. Also, we suggest adjusting the bibliography with a single citation style (e.g. references 12, 16, 23, etc.).
- Ref 16 please add: accessed on… with the date
Author Response
REVIEWER 3
|
||
Line 45. Pathobiology:
|
a brief comment on the importance of kidney biopsy in diagnosing exactly the diabetic kidney disease would be done. |
Details on renal biopsy is already described in the next section (3). However, we added a new sentence in this section and highlighted. |
Line 222 paragraph 6. tubulary biomarkers:
|
Tubular biomarkers, it is not clear why immunoglobulins belong to this category and not to glomerular markers. the authors conclude that they may offer a more precise approach to care: given the expectations related to metabolomics, why are they not used in clinical practice given the impressive clinical, social and even economic impact of DKD? a comment from the authors?
|
We agree with this point & the paragraph on urine immunoglobulins is now moved to section glomerular biomarkers. Additional paragraphs on metabolomics are added in section 7. Metabolomics to address the comments from other reviewers (marked) |
Line 317
|
“genome-wide association studies (EWAS)” should be maybe “esome-wide association studies” |
We have now changed this to “Exome-wide association studies” |
Line 365 9. Current disease modifying approaches
|
the authors, in addition to the promising results offered by the new hypoglycemic drugs, in discussion section and not only, briefly in conclusion, cannot help but mention the importance of lifestyle, strict dietary control in order to reduce hyperglycemia and physical activity, in addition to quitting smoking
|
We now have added a new paragraph at the beginning of the section 9 to address this issue with an additional sentence in the conclusion section. |
Line 382 MRA: |
The paragraph could be further elaborated, better explaining the role of the drugs in light of the physiological regulatory mechanism. In addition, examples of drugs in this category could be included. |
To make the text more concise, we avoided a lot of description in every section. However, we added a couple of sentences as suggested at the end of this paragraph now (marked). |
Line 464, 10.paragraph. Clinical studies. |
It might be useful to add a table summarizing what is discussed in paragraph 10, summarizing the characteristics of clinical trials on emerging therapies in DKD, this would lighten the text a bit. |
We have only provided a snapshot summary about these trials and without some text it is difficult to mention all in a table. Therefore, we haven’t created a new table. Hope the reviewer will kindly agree with the same. |
|
Please, add references when you cites articles or studies. Also, we suggest adjusting the bibliography with a single citation style (e.g. references 12, 16, 23, etc.). |
Some of the points we mentioned are well-known facts and we haven’t added references for those points to restrict the number of citations. We have changed ref 12,16, & 23 and modified other citations to get uniformity. |
|
Ref 16 please add: accessed on… with the date
|
Ref 16 is on PubMed and the citation style is now changed. |
Round 2
Reviewer 1 Report
Comments and Suggestions for Authors
Dear Authors,
Thank you for your thorough and thoughtful revisions. I have carefully reviewed the updated manuscript, and I appreciate the effort you’ve made in addressing all the comments and suggestions provided in my initial review.
The modifications have significantly improved the clarity, structure, and scientific value of the paper. I am satisfied with the changes and have no further comments at this stage.
Wishing you all the best with the final steps toward publication.
Best regards,